# Effects of Annealing for Strength Enhancement of FDM 3D-Printed ABS Reinforced with Recycled Carbon Fiber

**DOI:** 10.3390/polym15143110

**Published:** 2023-07-21

**Authors:** Wonseok Seok, Euysik Jeon, Youngshin Kim

**Affiliations:** 1Department of Future Convergence Engineering, Kongju National University, Cheonan 31080, Republic of Korea; 201601632@smail.kongju.ac.kr; 2Graduate Program for Eco-Friendly Future Automotive Technology, Kongju National University, Cheonan 31080, Republic of Korea

**Keywords:** recycled carbon fiber, fused deposition modeling, annealing, heat treatment, mechanical properties

## Abstract

This study investigates the effect of annealing on the mechanical properties of fused deposition modeling (FDM) 3D-printed recycled carbon fiber (rCF)-reinforced composites. In this study, filaments for FDM 3D printers are self-fabricated from pure acrylonitrile butadiene styrene (ABS) and ABS reinforced with fiber content of 10 wt% and 20 wt% rCF. This study explores the tensile and flexural properties as a function of the annealing temperature and time for the three different fiber content values. In addition, dimensional measurements of the shape changes are performed to determine the suitability of applying annealing in practical manufacturing processes. The results show that annealing improves the mechanical properties by narrowing the voids between the beads, which occur during the FDM process, and by reducing the gaps between the fibers and polymer. Following annealing, the largest tensile and flexural strength improvements are 12.64% and 42.33%, respectively, for the 20 wt% rCF content samples. Moreover, compared with the pure ABS samples, the annealing effect improves the mechanical properties of the rCF-reinforced samples more effectively, and they have higher dimensional stability, indicating their suitability for annealing. These results are expected to expand the application fields of rCF and greatly increase the potential use of FDM-printed parts.

## 1. Introduction

Additive manufacturing (AM), alternatively referred to as three-dimensional (3D) printing, is an innovative technique for rapid prototyping. It allows for the sequential fabrication of physical 3D parts based on computer-aided design, employing a layer-by-layer construction approach [1,2]. AM technology is increasingly applied in a diverse range of fields, such as aerospace, automotives, biomedical applications, and architecture [3,4,5], due to its ability to build complex geometries, highly customized production, minimal material loss, and low production costs [6,7,8]. Widely used AM technologies include fused deposition modeling (FDM), selective laser sintering, layered object manufacturing, direct light processing, and selective laser melting [9,10,11,12]. Of these, FDM is preferred because of its low cost, design freedom, and ability to quickly create structures with complex geometries [13].

In FDM, a thermoplastic material heated in a heat block is melted and extruded through a nozzle onto a build plate, which is where the 3D model attaches during the printing process and serves as the base [14,15]. The beads comprising each layer are stacked on top of each other to create a part with a 3D shape. Owing to the characteristics of this printing process, when materials are printed layer-by-layer, the printed parts typically have many voids between the beads, which greatly reduce the strength of the manufactured part [16,17]. In addition, some beads are extruded through the nozzle, while some beads have already undergone extrusion and begin to cool. This difference in heating and cooling induces internal stresses and weakens the interfacial bonds between the beads, which in turn degrades their mechanical properties [18]. Moreover, FDM methods using thermoplastics as feedstocks yield mechanical properties that are not appropriate for high-performance engineering composites, owing to the characteristics of the material [19]. These limitations, such as reductions in strength and the degradation of the mechanical properties, restrict the widespread application of FDM technologies [20].

Continuous research is being conducted to enhance the mechanical performance of FDM-printed products. This involves the systematic optimization of the FDM printing parameters, such as the nozzle temperature, layer thickness, build temperature, printing speed, and raster angle [21,22]. Optimization of the major printing parameters has improved the quality and mechanical properties of FDM 3D-printed products; nevertheless, there remain issues regarding the interlayer bonding of FDM parts, residual stresses, and the inherently poor mechanical properties of thermoplastics. Another approach is to add carbon fibers to a plastic material to form a thermoplastic matrix carbon fiber-reinforced plastic (CFRP) composite [23,24]. Ning et al. [14] compared FDM 3D-printed pure plastic specimens with those with 5 wt% CF content; the CF-reinforced specimens showed 11.82%, 16.82%, and 21.86% increases in flexural strength, flexural modulus, and flexural toughness, respectively. Heidari-Rarani et al. [25] fabricated continuous CF-reinforced thermoplastics using FDM 3D printers through an innovative extruder design and manufacturing. The bending and tensile strengths of the continuous CF-reinforced polylactic acid (PLA) composites increased by 108% and 35%, respectively, compared to pure PLA. These studies demonstrate that CFs can be used to efficiently modify the performance of 3D parts. However, the production of virgin CFs (vCFs) is energy-intensive and produces carbon dioxide [26]. In addition, most vCF-reinforced plastics are currently landfilled or incinerated at the end of their life [27,28]. Therefore, the use of vCFs has a negative environmental impact and causes economic losses owing to the disposal of high-value CFs.

As a sustainable and cost-effective alternative to vCFs, studies have been conducted on recycled CFs (rCFs), which have gained considerable attention owing to their environmental and economic benefits. Recovered rCFs are known to have approximately 70–90% of the strength of vCFs [29,30]. Liu et al. [31] showed that rCFs could effectively modify the mechanical strength, electrical conductivity, thermal conductivity, and wear resistance of parts produced by FDM 3D printing. However, although many studies have demonstrated that vCF and rCF reinforcements can improve the poor mechanical properties of thermoplastics, the problems of interlayer bonding and voids, which are fundamental problems in FDM 3D printers, remain a challenge to be solved. For example, owing to the increased pores caused by the addition of CFs, and their high thermal conductivity, the adhesion between layers is weakened and residual stresses occur [32,33]. Therefore, it is expected that higher mechanical strength can be achieved by reducing the residual thermal stresses inside rCF composites produced using FDM 3D printers, and by addressing the fundamental issues of porosity and interlayer bonding.

Annealing has long been applied as a post-processing method to improve materials’ mechanical properties. The annealing of polymers is known to improve their mechanical properties by enhancing interlayer bonding [34]. The annealing process involves heating the polymer to a higher temperature than its glass transition temperature (Tg) to increase the molecular mobility of the polymer chains, allowing them to adapt to this temperature for a specified amount of time. This is followed by a slow cooling process to room temperature [35]. Thus, the reflow increases the crystallinity of the material and improves its mechanical properties. Various studies have used annealing to achieve interlayer bonding and porosity in FDM 3D-printed parts. Akhoundi et al. [36] indicated that heat treatment increases the density of 3D-printed PLA parts, which improves interfacial bonding. Basgul et al. [37] found that heat-treating polyether ether ketone affected its porosity and mechanical properties. Kumar et al. [38] observed notable enhancements in the hardness, tensile strength, flexural strength, and impact strength of annealed PETG specimens, with improvements of 7.8%, 8.5%, 9.4%, and 5.5%, respectively. Moreover, annealed PETG+CF specimens exhibited even more substantial improvements, with increases of 14.8%, 22%, 10.5%, and 12.1% in the same mechanical properties. The presence of CF during annealing demonstrated superior strength gains in comparison to annealing without CF, which was attributed to the augmented interlayer diffusion bonding between the CF and polymer. Various studies have confirmed that annealing significantly improves the mechanical properties of composites produced by FDM 3D printing; however, research on the annealing of rCF-reinforced composites is limited.

The objective of this study was to demonstrate the effect of annealing on minimizing defects by improving bead and interlayer bonding to improve the mechanical properties of FDM parts reinforced with rCFs. In this study, the mechanical and morphological properties of ABS composites with different rCF content were analyzed after annealing. The purpose of this study was to investigate the reinforcement effect of increasing the rCF content and whether the problems of increased porosity and weakened interlayer bonds can be improved by annealing. For this purpose, we considered acrylonitrile butadiene styrene (ABS) without reinforcement, and fiber-reinforced ABS with 10 wt% and 20 wt% rCF content. To produce our filament for FDM printing, the filament extrusion temperature parameters were analyzed. The effects of annealing on the mechanical properties according to the temperature and time were analyzed. The temperature conditions were determined to be 105 °C, 125 °C, and 175 °C, which were higher than the Tg values, and exposure times of 0.5 h, 2 h, and 4 h were considered. Geometric changes induced by the annealing of FDM 3D-printed rCF-reinforced composites were observed, and the dimensional changes in the samples before and after annealing were measured. Furthermore, the effects of the different annealing conditions on the mechanical properties with respect to rCF content were analyzed. The microstructural changes induced by annealing were also analyzed using morphological analysis.

## 2. Materials and Methods

### 2.1. Filament Fabrication

Regarding rCF, little is known about the filament fabrication process for use in FDM 3D printers because the material is not widely used. Therefore, this study used rCF to self-fabricate filaments for FDM 3D printers and first set the process parameters. The material used was ABS, which is the most popular material used in FDM 3D printing processes because of its stable mechanical properties, good impact resistance and strength, and high thermal deformation temperature. The ABS utilized in this study was ABS HF380, manufactured by LG Chem, Ltd. (Seoul, Republic of Korea). The melt flow rate was 42 g/min by ASTM D1238. ABS HF380 consists of 25 wt% acrylonitrile, 16 wt% butadiene, and 59 wt% styrene. The rCF was obtained by procuring pellets (Cateck-h, Republic of Korea) that had different CF content (pure ABS, 10 wt% rCF, and 20 wt% rCF).

A single-screw extruder (3D Evo Composer 450, 3D Evo B.V., Utrecht, The Netherlands) was used to fabricate the rCF/ABS filaments. It consisted of four independently controllable heating barrels, a built-in winding system, and an optical sensor that monitored and controlled the diameter of the filament in real time, allowing filaments with a set diameter to be fabricated. The material was dried at 60 °C for 4 h prior to extrusion (manufacturer’s recommended temperature and time). The dried pellets were fed into the extruder to produce filaments with diameters of 1.75 mm and 2.85 mm, which are applicable to commercial FDM 3D printers. In addition, to determine the filament extruder temperature parameters, extrusion was performed under three heat-barrel temperatures, as listed in Table 1. The heat-barrel temperatures were within the range of 200–250 °C, which was the processing temperature recommended by the manufacturer. The other extruder parameters were experimentally determined to obtain stable surface quality and filament diameters. A schematic of the filament extruder and the location of each heat zone is shown in Figure 1.

### 2.2. FDM 3D Printing Specifications

All samples were produced using a FUNMAT HT FDM 3D printer (Intamsys Funmat HT, Intamsys, Ostfildern, Germany) and a BCN3D Epsilon W50 printer (BCN3D Technologies, Barcelona, Spain). The tensile and dimension analysis specimens were printed by the Intamsys 3D printer using the following printing parameters: (i) 20 mm/s printing speed; (ii) nozzle, bed, and chamber temperatures of 250 °C, 90 °C, and 65 °C, respectively; (iii) 100% infill density; (iv) line infill pattern; and (v) layer height of 0.2 mm. The flexural specimens were printed by the BCN3D Epsilon W50 printer using the following printing parameters: (i) 30 mm/s printing speed; (ii) nozzle and bed temperatures of 250 °C and 80 °C, respectively; (iii) 100% infill density; (iv) line infill pattern; and (v) layer height of 0.2 mm.

### 2.3. Annealing

After printing, the specimens were annealed at temperatures above the commonly known Tg values of ABS materials [39]. The annealing conditions applied throughout this study are shown in Table 2; three different temperatures (105 °C, 125 °C, and 175 °C) were considered, and the heating times were 0.5 h, 2 h, and 4 h. The exception was the temperature condition of 175 °C, where a time of only 0.5 h was considered due to extreme dimensional deformation.

The samples were annealed in a convection smart oven (Smart Gravity Air Drying Oven, Daehan Science, Wonju-si, Republic of Korea). Convection ovens provide a heat source with 360° heated air circulation, which allows for more uniform results compared to other heat sources. After heating under the specified conditions, the oven was not opened until the sample had cooled to room temperature, allowing gradual cooling. The process from FDM 3D printing to annealing is schematically illustrated in Figure 2.

### 2.4. Mechanical Testing

#### 2.4.1. Filament Single Fiber Tensile Testing

Single fiber tensile testing was conducted to observe the tensile properties of the filaments. A total of 13 temperature combinations were used to determine the filament extruder temperature parameters. For the test, the filament was cut into lengths of 20 cm. Tensile tests were performed using a SHIMADZU AG-X testing machine (SHIMADZU, Kyoto, Japan) with a load cell of 1 kN. The samples were tested at a constant crosshead displacement rate of 2 mm/min.

#### 2.4.2. FDM 3D-Printed Sample Mechanical Testing

Tensile and bending tests were conducted to observe the variations in the ultimate strength and modulus of elasticity with the heat treatment conditions and fiber content. Tensile specimens were prepared according to the ASTM D638 standard [40], and flexural specimens were prepared according to ASTM D790 [41]. The detailed dimensions of the samples are shown in Figure 3. Mechanical tests were performed using a Shimadzu AG-X testing machine with a 50 kN load cell. The specimens were tested at a constant crosshead displacement rate of 2 mm/min. To ensure repeatability, at least three samples were tested for each condition. The tensile and flexural strengths were calculated from the original cross-sectional area of the part from the maximum load during the test. The modulus of elasticity was calculated from the slope of the stress–strain curve within the elastic cross-section.

### 2.5. Dimensional Analysis

Thermoplastic parts produced by FDM are prone to structural deformation due to heat. These dimensional deformations must be considered when applying annealing. In particular, due to the unique nature of FDM parts, different variations in deformation are expected along the X, Y, and Z axes depending on the printing orientation. Therefore, it is important to evaluate the dimensional change of an annealed FDM 3D-printed sample compared to its initial state. To record the dimensional changes in the specimens, the lengths along the three axes (X, Y, and Z) of the treated specimens were measured using a digital Vernier caliper. To ensure statistical accuracy, the results were taken as the averages of three measurements for each axis. The detailed dimensions of the specimen used for the dimensional stability measurements are shown in Figure 3, and the coordinate axis settings of the sample are shown in Figure 4.

### 2.6. Morphology Analysis

The microstructures of the filament cross-sections with different rCF content and the fracture surfaces of the specimens as a function of the annealing temperature and time were investigated by scanning electron microscopy (SEM, TESCAN VEGA3, Brno, Czech Republic) to observe the internal defects in the fabricated filaments and to analyze the interlayer structure changes with the annealing of FDM 3D-printed specimens. This allowed us to observe the voids with increasing fiber content in the filaments, analyze the bonding between the regenerated CFs and polymers in the FDM-printed samples using the filament cross-sections, and determine the changes in the interlayer spacing and internal voids with the annealing conditions. All samples had Au/Pd coatings on the surface to improve the image resolution.

## 3. Characterization of Self-Fabricated rCF/ABS Filaments

### 3.1. Filament Single Fiber Tensile Test

Figure 5 shows the results of the single fiber tensile test of the ABS filament under different extrusion temperature conditions (Table 1). This process is essential in determining the filament extrusion process conditions.

In the single fiber tensile test results for each extruder temperature combination, the highest tensile strength is commonly found at the lowest temperature of 200 °C in heat zone 1, close to the nozzle. On the other hand, the lowest mechanical properties are found at the highest temperature of 250 °C in heat zone 1. The maximum and minimum ultimate tensile strength values are 35.39 MPa and 27.92 MPa, respectively, representing a strength difference of 26.74%. The modulus shows results similar to those of the tensile strength. The maximum and minimum moduli are found at 200 °C and 250 °C, respectively, in heat zone 1. The maximum and minimum modulus values are 3.69 GPa and 2.85 GPa, respectively, showing a difference of approximately 28.81%. This indicates that the mechanical properties of the filament are highest when the temperature of heat zone 1, close to the nozzle, is 200 °C, which is at the lower end of the processing temperature range. Additionally, the mechanical properties tend to decrease as the filament extruded at a high temperature is rapidly cooled upon meeting the outside air. However, there is no noticeable difference in heat zones 2–3 compared to heat zone 1. The final filament extrusion process parameters were based on the results of the single fiber tensile test, and the filaments used throughout this study were made according to the parameters shown in Table 3.

### 3.2. Analysis of Filament Fracture Surface

Cross-sectional analysis of the filaments by SEM was performed to check the build quality and defects of the filaments at different fiber content. Figure 6 shows the SEM images. As suggested in previous studies [32,33], the filaments exhibit increasing porosity with increasing rCF content. The voids observed in the 20 wt% rCF filament could have a negative impact on the mechanical properties. In addition, some gaps are observed between the fibers and the polymer. This indicates a weak interfacial bond between the fiber and polymer, which can lead to a decrease in the reinforcement effect of the fiber.

## 4. Results and Discussion of Annealed FDM 3D-Printed Specimens

### 4.1. Mechanical Properties

#### 4.1.1. FDM 3D-Printed Samples’ Tensile Test Results

First, a comparative study of three different rCF content values before annealing was conducted to determine the reinforcement effect of the rCF content on the tensile strength of the FDM-produced specimens. The tensile test results of the three types, namely pure ABS, ABS with 10 wt% rCF, and ABS with 20 wt% rCF, are shown in Figure 7.

The effect of the rCF content on the tensile strength is shown in Figure 7a. As the rCF content increases from 0 wt% to 10 wt%, the tensile strength improves, and the respective tensile strength averages are 24.7 MPa and 26.33 MPa, an increase of approximately 6.44%. Similarly, in the case of 10 wt% to 20 wt%, the tensile strength increases from 26.33 MPa to 29.59 MPa, representing an improvement of approximately 12.39%. The 20 wt% rCF/ABS, which shows the largest tensile strength, exhibits a tensile strength improvement of approximately 19.63% compared to pure ABS, which shows the lowest tensile strength. Figure 7b also shows that the modulus improves with increasing rCF content. The tensile modulus of pure ABS is 2.14 GPa, and the tensile moduli of the 10 wt% and 20 wt% rCF samples are 28.4 GPa and 3.35 GPa, respectively. Compared with the modulus of pure ABS, the moduli of the 10 wt% and 20 wt% rCF-reinforced samples increase by 31.94% and 61.57%, respectively. As shown in Figure 7c, the elongation at break decreases with increasing rCF content. This indicates that the material becomes brittle with the addition of recycled fibers.

Figure 8 shows the tensile test results at different annealing temperatures and times for different rCF content. The tensile strength of un-annealed pure ABS is 24.7 MPa. As shown in Figure 8a, when pure ABS is annealed at 105 °C, 125 °C, and 175 °C for the same time of 0.5 h, the tensile strengths are 24.7 MPa, 24.4 MPa, and 25.57 MPa, respectively. The tensile strengths at 105 °C tend to increase with increasing time. The tensile strengths at 2 h and 4 h are 26.05 MPa and 26.34 MPa, respectively. Pure ABS shows no change in tensile strength at 0.5 h at the annealing temperature of 105 °C; however, the tensile strengths after 2 and 4 h are higher than those of the untreated sample. At 125 °C and heating times of 0.5 h, 2 h, and 4 h, the tensile strengths are 24.4 MPa, 25.71 MPa, and 24.01 MPa, respectively. Only the 2 h time shows an improvement in strength, whereas the other times show a slight decrease. The maximum tensile strength of annealed pure ABS is observed at 105 °C and 4 h, showing an improvement of 6.31% compared to the result before annealing; the minimum tensile strength is observed at 125 °C and 4 h, showing a strength decrease of 2.94% compared to the sample before annealing. This is because of the residual stress caused by the shape deformation due to annealing of the tensile specimen at 125 °C (refer to Figure A4 in Appendix A).

Figure 8b shows the tensile strength results at different annealing temperatures and times for the ABS samples with 10 wt% rCF content. Compared with the un-annealed 10 wt% rCF/ABS with a tensile strength of 26.33 MPa, all annealing conditions show an improvement in tensile strength. The maximum tensile strengths are 28.49 MPa and 28.36 Mpa for the 2 h and 4 h heating times, respectively, at 105 °C. At 105 °C and 2 h heating time, the tensile strength shows an increase of 8.19% compared to that before treatment. On the other hand, the minimum tensile strength is 26.34 MPa at 125 °C and 4 h, and similar strengths of 26.6 MPa and 26.36 MPa are recorded at 0.5 h and 2 h, respectively, at 125 °C. This is almost equal to that before annealing. At the same time condition of 0.5 h, the results for the different temperature conditions of 105 °C, 125 °C, and 175 °C are 26.64 MPa, 26.6 MPa, and 27.61 MPa, respectively, showing an increase with temperature.

Figure 8c shows the tensile strength results versus annealing temperature and time for the ABS sample with 20 wt% rCF content. The tensile strength of the 20 wt% rCF/ABS sample before annealing was 29.59 MPa. The greatest tensile strength of 33.42 MPa is recorded at the 105 °C, 4 h condition, with an increase of approximately 12.94% over the untreated sample. The lowest tensile strength of 29.78 MPa is recorded at 175 °C and 0.5 h, indicating an improvement of only 0.62%. At a 0.5 h treatment time, the tensile strengths at the temperature conditions of 105 °C, 125 °C, and 175 °C are 32.07 MPa, 31.36 MPa, and 29.77 MPa, respectively. Thus, the tensile strength decreases with temperature. This is contrary to the trend of the pure ABS and 10 wt% rCF samples, where the strength generally increases with increasing temperature. This is believed to be due to the synergistic effect of the increased brittle behavior with the increase in the rCF content; the material becomes brittle owing to annealing and therefore fractures at small deformations.

Figure 9 shows the tensile modulus results for the annealed samples with different fiber content. The increase in the tensile modulus is more noticeable than that in the tensile strength. For all rCF content values, annealing improves the tensile modulus. The pure ABS has an elastic modulus of 2.16 GPa before treatment, and the highest elastic modulus of 2.56 GPa is obtained at 125 °C and 4 h, an improvement of approximately 19.12%. Compared to the case before annealing, the 10 wt% and 20 wt% rCF samples show the highest elastic moduli of 3.56 GPa and 4.58 GPa, respectively, at 175 °C and 0.5 h, improvements of 25.21% and 36.44%, respectively. The tensile modulus generally increases with increasing CF content, annealing treatment time, and temperature.

#### 4.1.2. FDM 3D-Printed Samples’ Flexural Test Results

The flexural properties of the FDM 3D-printed pure ABS and rCF-reinforced samples before annealing are shown in Figure 10. The flexural strength and modulus of the 10 wt% rCF/ABS sample are 61.36 MPa and 2.48 GPa, respectively, which is a 7.99% improvement in flexural strength and 23.57% improvement in modulus compared to the pure ABS sample (56.82 MPa and 2.01 GPa, respectively). The modulus and flexural strength of the 20 wt% rCF/ABS sample are 2.23 GPa and 39.73 MPa, respectively, showing an 11.06% modulus improvement over pure ABS, but a 30.08% decrease in flexural strength. As shown by the SEM analysis of the filaments (Figure 6), the 20 wt% samples exhibit significant porosity, and the strength reduction in materials with higher fiber content can be explained by weaker interlayer bonding [14,42].

The flexural properties of the samples after annealing are presented in Figure 11. The flexural properties show improvements in strength and modulus owing to the improved interlayer bonding and reduced porosity caused by annealing. The flexural strength of the pure ABS sample prior to annealing is 56.82 MPa. As shown in Figure 11a, the flexural strength of pure ABS shows the greatest result of 66.22 MPa at 175 °C and 0.5 h, indicating an improvement of 16.54%. Additionally, the lowest flexural strength of 59.62 MPa after annealing is found at 105 °C and 0.5 h, an improvement of 4.93% compared to that before treatment. The 10 wt% rCF/ABS has the maximum flexural strength of 61.36 MPa before annealing. As shown in Figure 11b, the maximum flexural strength of 70.61 MPa is recorded after annealing at 175 °C and 0.5 h. Compared to the case before annealing, the flexural strength is improved by 15.08%. The lowest flexural strength for the 10 wt% rCF/ABS sample (63.22 MPa) is recorded at 125 °C and 0.5 h; nevertheless, it is still 3.02% higher than that before treatment. The 10 wt% rCF sample is observed to warp at 125 °C, which may explain the lower results compared to the other temperatures. The 20 wt% rCF/ABS sample shows somewhat lower strength before treatment, owing to its higher porosity compared to the other samples. However, as shown in Figure 11c, the flexural strength improves by 42.33% to 56.65 MPa after annealing at the 175 °C, 0.5 h condition, which is the largest improvement compared to other fiber content values. This indicates that many of the existing pores are healed by annealing, and the annealing effect is most obvious at this level of rCF content.

The flexural moduli of the annealed samples are shown in Figure 12. The modulus shows a mostly increasing trend with increasing annealing temperature, similar to the trend with increasing processing time. Similar to the flexural strength results, the largest modulus values are observed at the highest processing temperature of 175 °C, which shows that the modulus increase eventually correlates with the processing temperature. At 175 °C and 0.5 h, pure ABS shows a 17.69% increase in modulus, from 2.01 GPa pre-treatment to 2.37 GPa post-treatment, while the 10 wt% sample shows a 34.32% improvement, from 2.48 GPa pre-treatment to 3.34 GPa post-treatment. The 20 wt% sample shows the highest improvement in flexural strength, with a 71.47% improvement from 2.23 GPa pre-treatment to 3.83 GPa post-treatment.

### 4.2. Post-Annealing Dimension Analysis

Figure 13 shows the relative change values for the three axes (Figure 4) of pure ABS, 10 wt% rCF/ABS, and 20 wt% rCF/ABS samples under different annealing conditions. Figure 13a shows the rate of dimensional change in the *x*-axis direction, which is also the printing direction. The dimensional change along the *x*-axis does not show much shrinkage at the relatively low temperature of 105 °C. However, as the annealing temperature and processing time increase, the samples show significant shrinkage at 125 °C and 175 °C. The most noticeable shrinkage is recorded for pure ABS, with maximum shrinkage of −27%. On the other hand, the maximum shrinkages at 10 wt% and 20 wt% rCF are −18.8% and −13.87%, respectively, and the variation in these shrinkage rates decreases as the fiber content increases. Figure 13b,c show the relative change rates of the *y*-axis direction and *z*-axis (thickness direction), respectively, which tend to expand with increasing temperature and treatment time, as opposed to the shrinkage observed in the *x*-axis. The dimensional change rate for the *y*-axis was relatively small compared to those for the X and Z-axes, and no significant change was observed for the 20 wt% rCF sample. This was also seen in the actual tensile specimens, as shown in Figure A1, Figure A2 and Figure A3 in Appendix A. This behavior indicates that the dimensional stability of the rCF is increased by annealing.

### 4.3. Post-Annealing Morphorlogy Analysis

The contrast between the sample before annealing and that at the highest temperature of 175 °C is shown in Figure 14. Owing to the nature of the printing process, FDM 3D-printed samples are created by extruding the beads that make up each layer through a nozzle. This creates gaps between the beads. The gaps between the beads increase the porosity of the printed product, resulting in a loss of mechanical properties. Before annealing, the sample exhibits inter-bead gaps. However, at 175 °C, the highest temperature applied in this study, the gaps between the beads are healed. This gap healing results in improved interlaminar bonding of the FDM parts. These results are consistent with those of the flexural tests in Section 4.1.2.

Generally, rCF is used to improve the poor mechanical properties of thermoplastics; however, if the interfacial bond between the polymer and fiber is inferior, the reinforcement effect of the fiber will be reduced because of the pull-out phenomenon at loads lower than the effective load. Therefore, the mechanical properties of the material can only be expressed when the bond between the rCF and polymer is complete. However, without post-treatment, the bond between the rCF and polymer is poor. Nevertheless, if annealing is applied, the gap between the fiber and polymer can be closed owing to polymer shrinkage. This phenomenon was confirmed by analyzing the fracture surface before and after the annealing of the sample with rCF added (20 wt% rCF/ABS), as shown in Figure 15. The un-annealed sample shows a gap between the rCF and ABS, as well as traces of CF pull-out around the periphery, indicating poor interfacial bonding between the fiber and polymer. However, the annealed sample shows that the shrinkage of the polymer eliminated the gap between it and the fiber, allowing the polymer to completely wrap the fiber. This characteristic has a significant impact on the improvement of the mechanical properties of fiber-reinforced plastics, which is consistent with the greater strength improvement of the rCF-reinforced sample than that of pure ABS.

## 5. Conclusions

This study investigated the effects of rCF reinforcement and annealing on the mechanical properties of FDM 3D-printed thermoplastics. The filaments used for FDM were self-fabricated to have different rCF content. Annealing improved the strength and quality of the FDM 3D-printed rCFRPs. Consequently, the porosity of the material caused by the addition of rCF was reduced, the gaps between the beads that occurred during the FDM 3D printing process were minimized, and the interlayer bonding was improved. It has been confirmed that annealing expands the technical potential of recycled carbon fibers as fillers. Consequently, it was possible to ascertain their substantial viability as a substitute for vCF, which is a material with significant environmental concerns. Moreover, this work is believed to expand the potential for FDM-produced parts to be utilized as real-world components rather than merely prototypes. The following can be concluded from this study.

The mechanical properties of the filament produced under various temperature combinations of the filament extruder were found to be closely related to the temperature of heat zone 1. The conditions that showed the highest tensile properties were all achieved when the temperature of heat zone 1, which was close to the nozzle, was the lowest at 200 °C. However, heat zones 2–3 did not significantly affect the tensile properties.Cross-sectional microstructure analysis of filaments with different fiber content showed that internal pores increase with increasing fiber content; in particular, large and small pores were observed at 20 wt% rCF, suggesting that internal pores increase with increasing fiber content, which may adversely affect the mechanical properties.Tensile and flexural tests showed that the addition of rCF significantly affected the mechanical properties. The tensile strength and modulus increased with fiber reinforcement, and the flexural modulus increased with the fiber content. The exception was the flexural strength at 20 wt% rCF, which was lower than that of pure ABS. This decrease in flexural strength is consistent with the defects observed in the filament cross-sectional analysis with increasing fiber content.Most of the annealed specimens had their highest tensile strengths at 105 °C and 4 h, and the tensile modulus increased proportionally with increasing treatment temperature and time. The largest increase was at 20 wt% rCF, where the strength and modulus increased by 12.94% and 36.44%, respectively.The flexural properties showed an increase in strength and modulus due to annealing at all temperature and time conditions. The largest increase in flexural stress (42.33%) and modulus (71.47%) occurred at 175 °C for the 20 wt% rCF sample.Observation of the fracture surfaces of the tensile sample after annealing confirmed that the voids between the beads caused by the FDM process healed as the annealing temperature increased. We also observed a reduction in micropores due to the addition of rCF and the closure of the gaps between the rCF and polymer. This was closely related to the improvement in the mechanical properties due to annealing.Annealing has a significant impact on sample dimensions. In particular, it causes large shrinkage in the same direction as the printing direction in FDM printing. However, this deformation decreases significantly as the rCF content increases.The results of the mechanical property improvement and dimensional stability due to annealing show that rCF-reinforced ABS is more suitable for annealing than pure ABS.

## Figures and Tables

**Figure 1 polymers-15-03110-f001:**
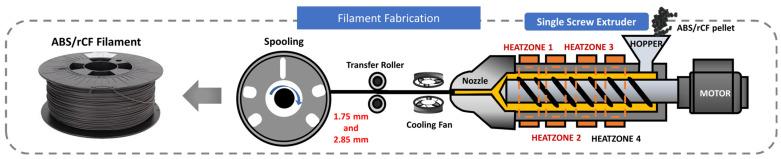
Schematic diagram of filament fabrication process.

**Figure 2 polymers-15-03110-f002:**
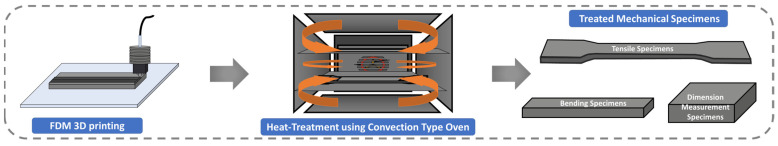
Schematic diagram from FDM 3D printing to annealing.

**Figure 3 polymers-15-03110-f003:**
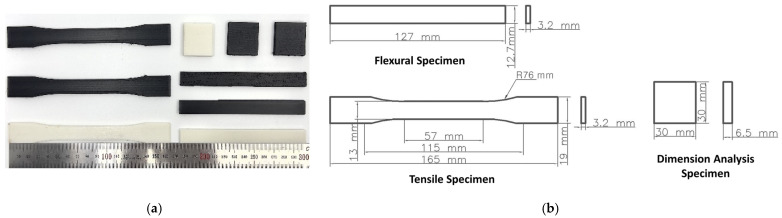
FDM 3D-printed tensile, flexural, and dimension analysis samples: (**a**) image of samples printed using FDM method; (**b**) detailed dimensions of machine test samples.

**Figure 4 polymers-15-03110-f004:**
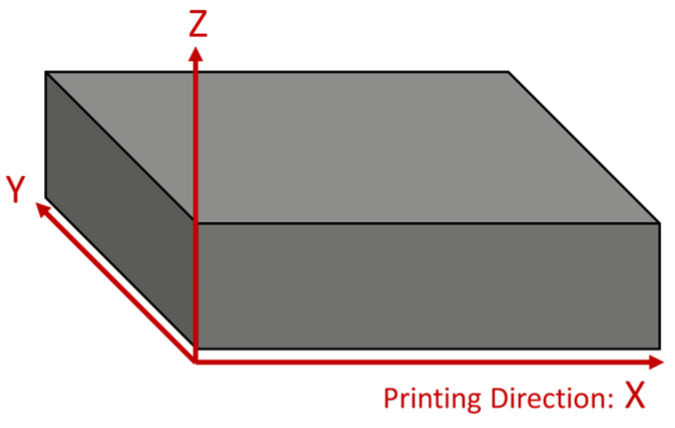
Coordinate axis setup for dimensional stability samples.

**Figure 5 polymers-15-03110-f005:**
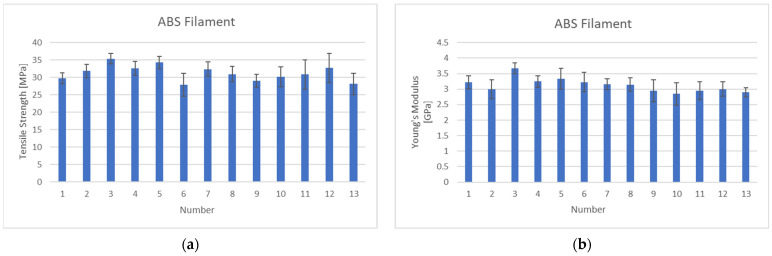
Filament single fiber tensile test results: (**a**) Tensile strength; (**b**) Young’s modulus.

**Figure 6 polymers-15-03110-f006:**
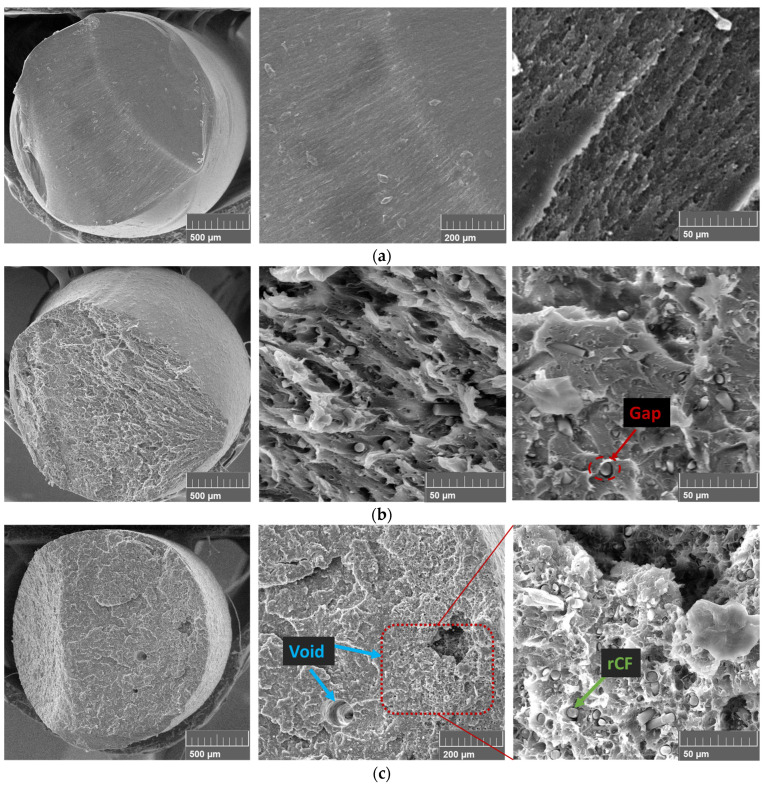
Filament fracture morphology: (**a**) pure ABS filament; (**b**) 10 wt% rCF/ABS filament; (**c**) 20 wt% rCF/ABS filament.

**Figure 7 polymers-15-03110-f007:**
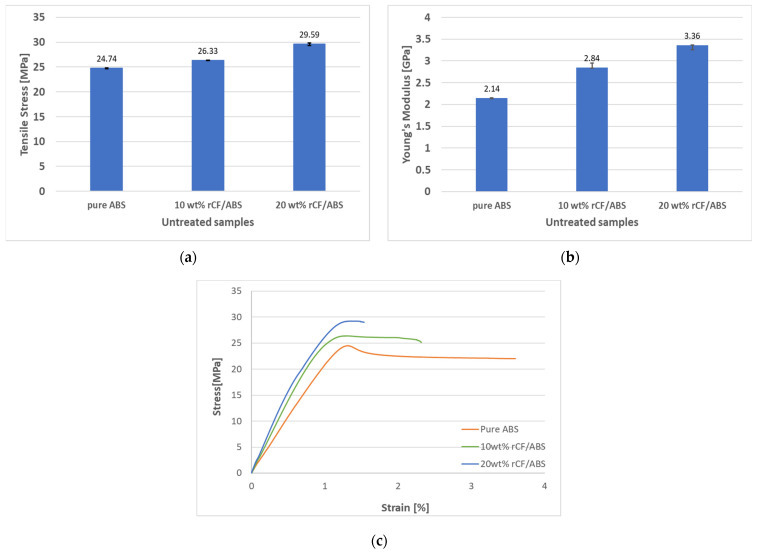
Tensile properties of pure ABS, 10 wt% rCF/ABS, and 20 wt% rCF/ABS samples before annealing: (**a**) tensile stress; (**b**) tensile modulus; (**c**) representative stress–strain curve.

**Figure 8 polymers-15-03110-f008:**
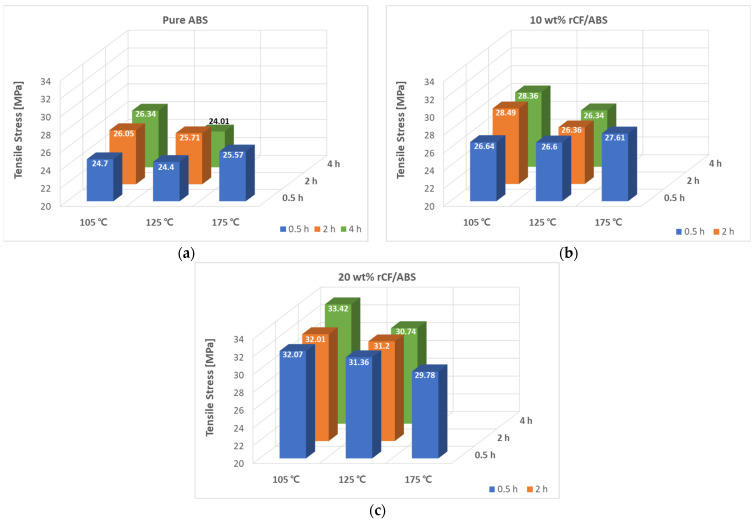
Tensile stress according to annealing: (**a**) annealed pure ABS; (**b**) annealed 10 wt% rCF/ABS; (**c**) annealed 20 wt% rCF/ABS.

**Figure 9 polymers-15-03110-f009:**
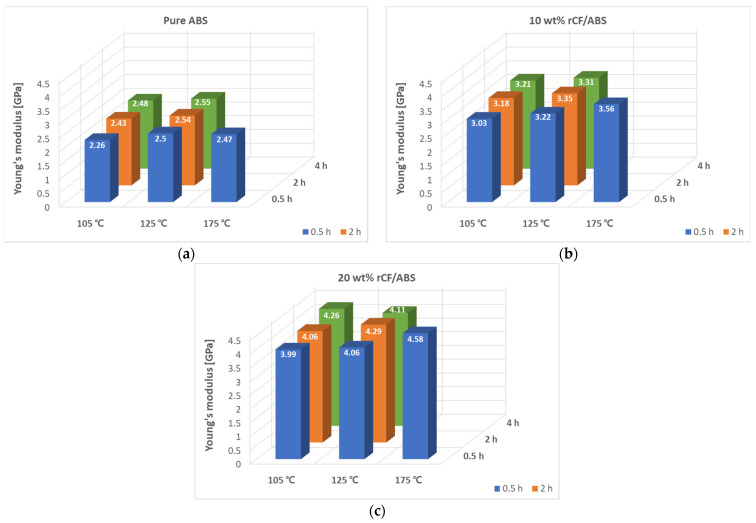
Tensile modulus according to annealing time and temperature: (**a**) annealed pure ABS; (**b**) annealed 10 wt% rCF/ABS; (**c**) annealed 20 wt% rCF/ABS.

**Figure 10 polymers-15-03110-f010:**
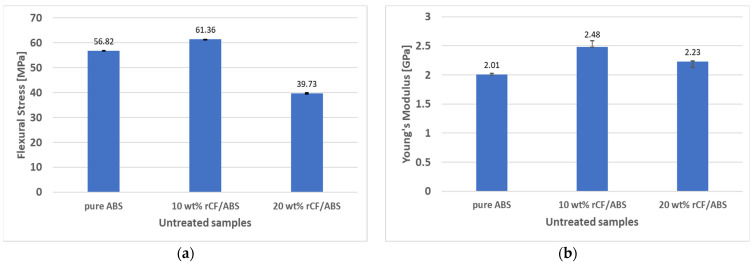
Flexural properties of pure ABS, 10 wt% rCF/ABS, and 20 wt% rCF/ABS before annealing: (**a**) flexural stress; (**b**) flexural modulus.

**Figure 11 polymers-15-03110-f011:**
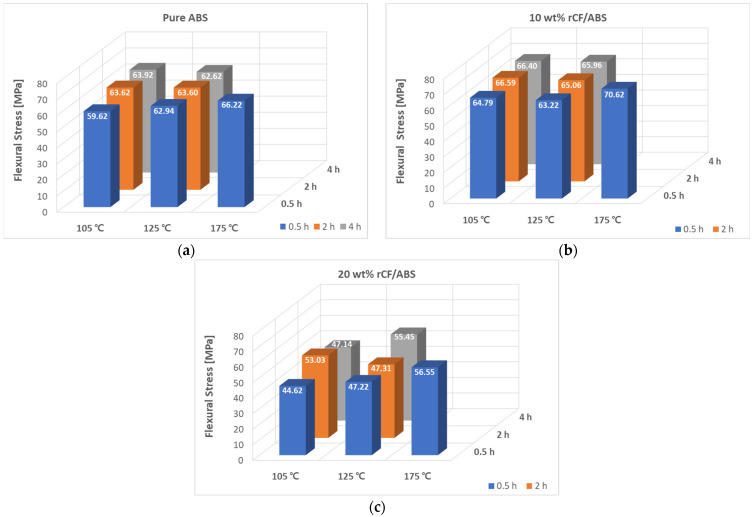
Flexural stress according to annealing time and temperature: (**a**) annealed pure ABS; (**b**) annealed 10 wt% rCF/ABS; (**c**) annealed 20 wt% rCF/ABS.

**Figure 12 polymers-15-03110-f012:**
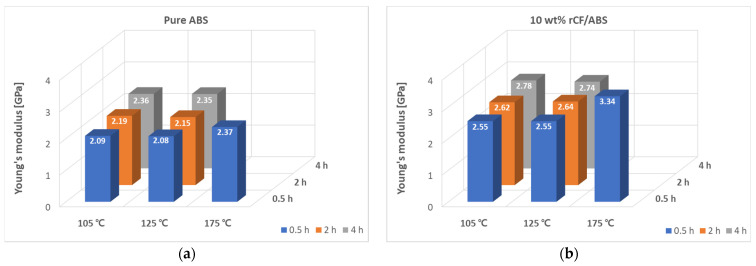
Flexural modulus according to annealing time and temperature: (**a**) annealed pure ABS; (**b**) annealed 10 wt% rCF/ABS; (**c**) annealed 20 wt% rCF/ABS.

**Figure 13 polymers-15-03110-f013:**
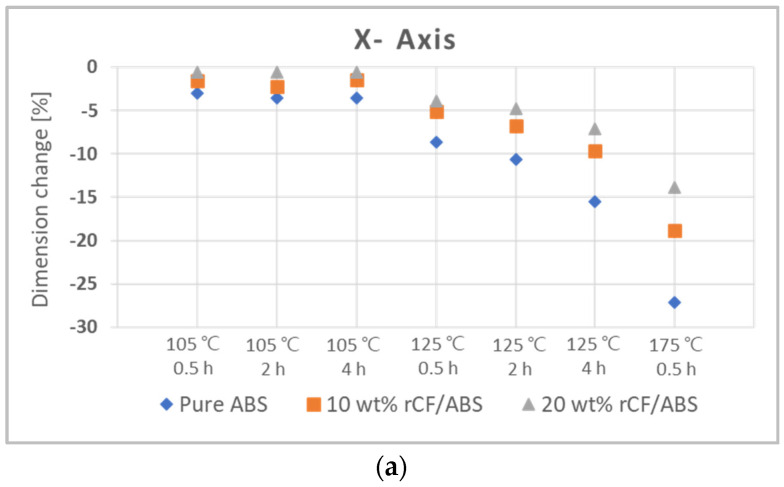
Relative changes in dimension for the annealed samples: (**a**) *X*-axis; (**b**) *Y*-axis; (**c**) *Z*-axis.

**Figure 14 polymers-15-03110-f014:**
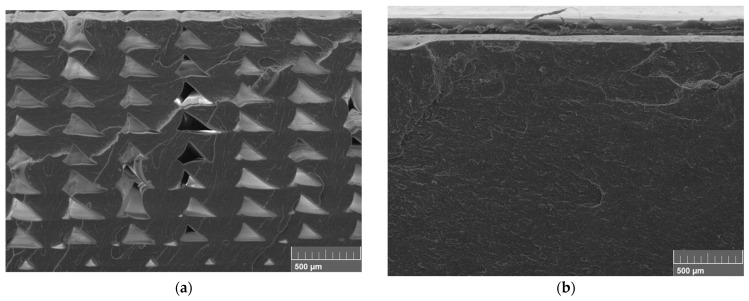
SEM image of tensile testing samples of before and after annealing: (**a**) un-annealed pure ABS sample; (**b**) pure ABS samples annealed at 175 °C, 0.5 h.

**Figure 15 polymers-15-03110-f015:**
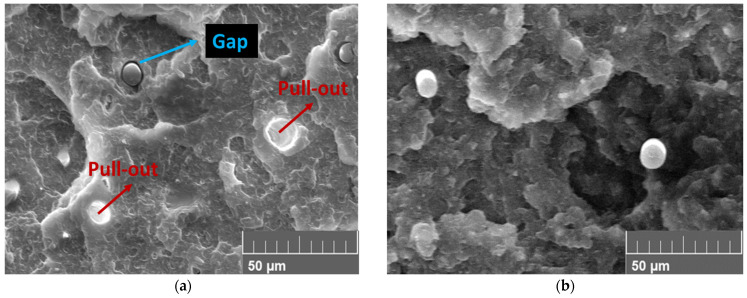
SEM images of 20 wt% rCF/ABS samples before and after annealing: (**a**) un-annealed sample; (**b**) sample annealed at 105 °C, 4 h.

**Table 1 polymers-15-03110-t001:** Temperature conditions for extruder parameter setting.

Temperature Combination	Heater 1 [°C]	Heater 2 [°C]	Heater 3 [°C]	Heater 4 [°C]
1	200	200	225	240
2	250	200	225	240
3	200	250	225	240
4	250	250	225	240
5	200	225	200	240
6	250	225	200	240
7	200	225	250	240
8	250	225	250	240
9	225	200	200	240
10	225	250	200	240
11	225	200	250	240
12	225	250	250	240
13	225	225	225	240

**Table 2 polymers-15-03110-t002:** Annealing parameters.

Number	Annealing Temperature (°C)	Time (h)
1	Untreated	-
2	105	0.5
3		2
4		4
5	125	0.5
6		2
7		4
8	175	0.5

**Table 3 polymers-15-03110-t003:** Final filament fabrication parameters.

Diameter [mm]	Heater 1 [°C]	Heater 2 [°C]	Heater 3 [°C]	Heater 4 [°C]	Extruder RPM	Fan Speed [%]
1.75	200	230	230	240	3.5	55
2.85					6.5	65

## Data Availability

Not applicable.

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
