# Peer review of "Effects of Annealing for Strength Enhancement of FDM 3D-Printed ABS Reinforced with Recycled Carbon Fiber"

_polymers, 2023, doi:10.3390/polym15143110_

Round 1

Reviewer 1 Report

Review of: Effects of Annealing for Strength Enhancement of FDM 3D Printed ABS Reinforced with Recycled Carbon Fiber

Authors: WonSeok Seok, EuySik Jeon, YoungShin Kim.

Overall, the work is presented well, with the authors providing a systematic investigation on the reinforcement effect of recycled carbon fibres and the annealing process parameters on the mechanical properties of 3D printed FDM based rCF-ABS composites. However, the manuscript needs some corrections before publication and overall improvement is needed in the grammar, especially in the framing of sentences and conveying of ideas.

·       Lines 13,14 - Rephrase/revise to have better choice of words for clarity and communication of idea, regarding choice of filaments.

·       Lines 17,18 - Give clarity to the usage of word ‘processes’. A more precise mention of application/function would provide more clarity.

·       Lines 42-Rephrase the description of the plate.

·       Lines 46—49 – Too long and grammatically incorrect-Rephrase/break down the sentence.

·       Lines 51-52- It would be appropriate to mention some of the limitations as a brief overview rather than mention the citation.

·       Lines 72 – Please insert a conclusive word as “Therefore”, “in other words” at the starting of the sentence.

·       Lines 97 – Rephrase to include the mention of rCF cases than just the PLA.

·       Lines 109- Rephase/revise the words “defect removal” as this is not achievable and the aim should be practical, viz. minimisation.

·       Lines 111-112– The sentence is incorrect, as properties are studied “after” annealing, rather than “by” annealing. Please improve sentence framing and usage of words for better communication.

·       Lines 121-125- Too long-Rephrase/break down the sentence.

· Lines 130-131- “Self-fabricate”- Better to explain the process of manufacturing.

·       Lines 185-Emphasize on the need of carrying out the single fibre test, by choosing right sentence formations to avoid confusion and improve clarity of ideas.

·       Lines-269-269-Rephrase and break down the statement. The idea that specimens analysed were before heat treatment should be highlighted.

·       Lines-296-297-can be rephrased to indicate that the mechanical property comparison is with respect to untreated specimens.

·       Lines 327-329-Rephrase/revise/breakdown line to communicate the idea.

· Lines 446—449-Breakdown/rephrase the sentence for better communication

·       Lines 450-451-Argument would be more impactful if some descriptions of relevant applications can be mentioned, rather than abstract mentions.

Comments on the quality of English language have been given as reviewer comments.

Author Response

Thank you for taking the time to review our paper. We greatly appreciate your insightful feedback and suggestions for improvement. We have carefully considered your comments and have made revisions accordingly. Please find the revised version attached with the necessary corrections incorporated.

Additionally, an expert was consulted to proofread and correct the grammar of the entire document.

Point 1: I Lines 13,14 - Rephrase/revise to have better choice of words for clarity and communication of idea, regarding choice of filaments.

Response 1: Regarding filament selection, we have modified the word to better select for clarity and idea delivery. The modifications are as follows. “In this study, filaments for FDM 3D printers are self-fabricated from pure acrylonitrile butadiene styrene (ABS), ABS reinforced with fiber content of 10 wt% and 20 wt% rCF.”[Lines 13-15]

Point 2: Give clarity to the usage of word ‘processes’. A more precise mention of application/function would provide more clarity.

Response 2: To clearly explain the use of the word 'process', we have rewritten the following sentences. “In addition, dimensional measurements of the shape changes were performed to determine the suitability of applying annealing in practical manufacturing processes.”[Lines 17-18]

Point 3: Rephrase the description of the plate.

Response 3: I wrote a more specific description of the build plate. “In FDM, a thermoplastic material heated in a heat block is melted and extruded through a nozzle onto a build plate, which is where the 3D model attaches during the printing process and serves as the base.”[Lines 43-44]

Point 4: Too long and grammatically incorrect-Rephrase/break down the sentence.

Response 4: We divided the paragraphs and modified the grammatical parts to reflect the "too long and grammatically incorrect" The reorganized sentence looks like this; “In addition, some beads are extruded through the nozzle, while some beads have al-ready undergone extrusion and begin to cool. This difference in heating and cooling induces internal stresses and weakens the interfacial bonds between the beads, which in turn degrades their mechanical properties.”[Lines 48-51]

Point 5: It would be appropriate to mention some of the limitations as a brief overview rather than mention the citation.

Response 5: This has undergone some modifications. “These limitations such as reduction in strength and degradation of mechanical properties restrict the widespread application of FDM technologies.”[Lines 54-55]

Point 6: Please insert a conclusive word as “Therefore”, “in other words” at the starting of the sentence.

Response 6: As you pointed out, I modified it to start with 'Therefore'.

“Therefore, the use of vCFs has a negative environmental impact and causes economic losses owing to the disposal of high-value CFs.”[Line 74]

Point 7: Rephrase to include the mention of rCF cases than just the PLA.

Response 7: Thank you for your good point. I also looked for references on rCF, but the research applying rCF was very limited, so I couldn't add it because there was no suitable reference.

Point 8: Rephase/revise the words “defect removal” as this is not achievable and the aim should be practical, viz. minimisation.

Response 8: It has been rewritten as follows to modify the change to the word you suggested. Modified from 'defect removal' to 'minimizing defects'. [Line 111]

Point 9: The sentence is incorrect, as properties are studied “after” annealing, rather than “by” annealing. Please improve sentence framing and usage of words for better communication.

Response 9: Modified from 'by annealing' to 'after annealing'. [Line 114]

Point 10: Too long-Rephrase/break down the sentence.

Response 10: We've broken up and reorganized the sentence to reflect your point about it being long.

“Geometric changes induced by the annealing of FDM 3D printed rCF-reinforced composites were observed, and dimensional changes in the samples before and after annealing were measured. Furthermore, the effects of different annealing conditions on the mechanical properties with respect to rCF content were analyzed.”[Lines 123-127]

Point 11: “Self-fabricate”- Better to explain the process of manufacturing.

Response 11: We've avoided major edits to avoid duplication, as the sentences about "Self-fabricate" include manufacturing processes in the immediately following sentences. If you have any comments on this, please let us know and we will incorporate them.

Point 12: Emphasize on the need of carrying out the single fibre test, by choosing right sentence formations to avoid confusion and improve clarity of ideas.

Response 12: To avoid confusion about the single fiber tensile test, and to improve clarity, the following modifications have been made.

“Single fiber tensile testing was conducted to observe the tensile properties of the filaments. A total of 13 temperature combinations were used to determine the filament extruder temperature parameters.”[Lines 190-192]

Point 13: can be rephrased to indicate that the mechanical property comparison is with respect to untreated specimens.

Response 13: To emphasize that the analyzed specimen was before heat treatment, the sentence was rephrased as follows; “First, a comparative study of three different rCF contents before annealing was con-ducted to determine the reinforcement effect of rCF content on the tensile strength of the FDM-produced specimens.” [Lines 273-275]

Point 14: Rephrase/revise/breakdown line to communicate the idea.

Response 14: The following modifications have been made to make the sentences more concise and more effective at conveying ideas.

This is believed to be due to the synergistic effect of the increased brittle behavior with the increase in the rCF content; the material becomes brittle owing to annealing and therefore fractures at small deformations.”[Lines 331-333]

Point 15: Breakdown/rephrase the sentence for better communication

Response 15: The following modifications have been made for better communication. Annealing improves the strength and quality of the FDM 3D printed rCFRPs. Consequently, the porosity of the material caused by the addition of rCF is reduced, the gaps between the beads that occur during the FDM 3D printing process are minimized, and interlayer bonding is improved.”[Lines 450-453]

Point 16: Argument would be more impactful if some descriptions of relevant applications can be mentioned, rather than abstract mentions.

Response 15: To reflect this, we have clarified the feasibility of recycled carbon fiber as a filler and its suitability as a vCF substitute. Therefore, the following sentence has been added. “It has been confirmed that annealing expands the technical potential of recycled carbon fibers as fillers. Consequently, it was possible to ascertain their substantial viability as a substitute for vCF, which is a material with significant environmental concerns.”

Reviewer 2 Report

Dear,

The authors prepared ABS composites with recycled carbon fiber. At the same time, the effect of annealing was investigated and how it affects the properties themselves. Manuscripts on material recycling are welcome, with a view to reintroducing them into the production chain. The manuscript has merit for publication, with some minor revisions:

> Materials. Please detail the ABS used, informing the melt flow index, and the proportion of each monomer;

> FDM 3D printed sample mechanical testing. Report how many samples were tested during the mechanical tests;
> Results and discussion. Please compare the results obtained with other reports in the literature;

> Conclusions. Conclude by reporting the importance of reusing carbon fiber for the environment and the technological potential as a filler;

Minor editing of English language required

Author Response

Thank you for taking the time to review our paper. We greatly appreciate your insightful feedback and suggestions for improvement. We have carefully considered your comments and have made revisions accordingly. Please find the revised version attached with the necessary corrections incorporated.

Additionally, a professional was consulted to grammatically correct the entire document.

Point 1: Materials. Please detail the ABS used, informing the melt flow index, and the proportion of each monomer.

Response 1: "Chapter 2: Materials and Methods, Section 2.1: Filament Preparation" has been revised to provide details on the ABS model name, melt flow index, and supply of each monomer used (lines 134-136). For the purposes of this thesis, this information has been included as follows:

“The ABS utilized in this study is ABS HF380, manufactured by LG Chem, Ltd. The melt flow rate was 42 g/min by ASTM D1238. ABS HF380 consists of 25 wt% acrylonitrile, 16 wt% butadiene, and 59 wt% styrene”

Point 2:  FDM 3D printed sample mechanical testing. Report how many samples were tested during the mechanical tests.

Response 2: Chapter 2: Materials and Methods, Section 2.4: FDM 3D Print Sample Mechanical Tests" (Line 201-202) stated that a minimum of three specimens were tested for each condition during the mechanical testing phase.

In this paper, “To ensure repeatability, at least three samples were tested for each condition.”

Point 3: Results and discussion. Please compare the results obtained with other reports in the literature. 

Response 3: Due to the nature of composite materials, results can vary greatly depending on the manufacturer of raw materials used as polymers, the mechanical properties of rCF as a filler, and process variables during manufacture. As a result, these characteristics were not individually compared. Therefore, the comparison in this study was inevitably based on the results of the annealing variable.

Point 4: Conclusions. Conclude by reporting the importance of reusing carbon fiber for the environment and the technological potential as a filler. 

Response 4: The following statements are added to indicate the environmental benefits of recycled carbon fiber and its technical potential as a filler :

It has been confirmed that annealing expands the technical potential of recycled carbon fibers as fillers. Consequently, it was possible to ascertain their substantial viability as a substitute for vCF, which is a material with significant environmental concerns.
